

# Ensemble modeling of stochastic unsteady open-channel flow in terms of its time-space evolutionary probability distribution: theoretical development

Alain Dib and M. Levent Kavvas

Department of Civil and Environmental Engineering, University of California, Davis, 95616, USA

*Correspondence to*: Alain Dib (aedib@ucdavis.edu)

**Abstract.** The Saint-Venant equations are commonly used as the governing equations to solve for modeling the spatially varied unsteady flow in open channels. The presence of uncertainties in the channel or flow parameters renders these
equations stochastic, thus requiring their solution in a stochastic framework in order to quantify the ensemble behavior and the variability of the process. While the Monte Carlo approach can be used for such a solution, its computational expense and its large number of simulations act to its disadvantage. This study proposes, explains, and derives a new methodology for solving the stochastic Saint-Venant equations in only one shot, without the need for a large number of simulations. The proposed methodology is derived by developing the nonlocal Lagrangian–Eulerian Fokker–Planck Equation of the
characteristic form of the stochastic Saint-Venant equations for an open-channel flow process, with an uncertain roughness coefficient. A numerical method for its solution is subsequently devised. The application and validation of this methodology are provided in a companion paper, in which the statistical results computed by the proposed methodology are compared against the results obtained by the Monte Carlo approach.

## 1 Introduction

Unsteady open-channel flows are a common occurrence in hydrology and hydraulics problems. They arise as a result of the movement of water waves in natural or artificial channels (Sturm, 2001). Understanding and tracing the movement of such water waves along the channels is of great importance in addressing engineering flow problems, including flood forecasting, flood control, hydrograph generation, and several others (Chow, 1959). The technique used to approximate and trace such water waves is known as flood routing, and the governing equations that are commonly used to solve for the unsteady flows
in flood routing problems are known as the Saint-Venant equations (Chanson, 2004).

      Various uncertainties can add to the complexity of solving the Saint-Venant equations in engineering routing problems (Gates and AlZahrani, 1996a; Ercan and Kavvas, 2012a). Such uncertainties may correspond to several factors. Physical conditions of open channels may be uncertain due to their high degree of variability (Sturm, 2001). One example is Manning's roughness coefficient, which greatly depends on the channel vegetation, brush, bed material, bedforms, and even



on the position of the free water surface (Chow, 1959; Sturm, 2001; Ercan and Kavvas, 2012a). With the uncertainties in quantifying or characterizing these factors, the roughness coefficient becomes extremely difficult to estimate (Sturm, 2001), rendering it uncertain. Channel geometric parameters may also be uncertain. This includes the channel bed slope (Ercan and Kavvas, 2012a) and the channel cross section geometry, the latter of which may exhibit significant spatial variability across a

river due to its irregular form and due to the changes it may undergo along the direction of flow (Chow, 1959). Other uncertainties may also arise from lateral inflows and initial conditions due to their spatial and/or temporal variability (Liang and Kavvas, 2008), as well as from the upstream boundary conditions due to the temporal variability of the inflows into the channel.

As a result of such uncertainties in the channel and flow parameters, these parameters may be considered spatially

and/or temporally random at the local scale of a river cross section, rendering the system behavior uncertain (Gates and AlZahrani, 1996a). Therefore, deterministically solving the Saint-Venant equations in this case would no longer be providing a representative solution to the flood routing problem being considered. In fact, in this study the governing partial differential equations (PDEs), i.e., the Saint-Venant equations, will be transformed into stochastic PDEs because the channel and flow parameters are now stochastic and can be described as random functions (Liang and Kavvas, 2008). This means that the

dependent variables that will be solved for by these equations (e.g., flow velocity and depth) will also be spatiotemporal random functions. Hence, instead of solving for the deterministic values of the dependent variables, the goal will be to solve for their statistical properties (Van Kampen, 1976), which can be obtained at designated discrete time-space positions.

Two popular methods can be used to estimate the statistical properties of the dependent variables for nonlinear problems with stochastic parameters: the finite-order analysis and the Monte Carlo (MC) approach. Applying the expectation

operator, or any other statistical moment operator, to nonlinear difference equations with stochastic parameters may result in nonlinear expressions that are difficult, or even impossible, to simplify into terms involving the known moments of the model parameters and the unknown moments of the dependent variables (Gates and AlZahrani, 1996a). Finite-order analysis overcomes this by performing a Taylor series expansion of the difference equations about the expected values of the parameters, from which higher-order terms are truncated. For example, truncating the Taylor series after the first-order term

is known as the first-order approximation, which is a good approximation when the system nonlinearity is not too high, and when the stochastic parameters have relatively small coefficients of variation (Dettinger and Wilson, 1981). However, with highly nonlinear problems, instead of using higher-order approximations of the finite-order method, it may be required and more efficient to use full-distribution methods such as the MC approach (Dettinger and Wilson, 1981).

The MC approach is well-known for simulating differential equations with stochastic parameters, and is used to

determine the distributions of the unknown stochastic dependent variables (Freeze, 1975; Smith and Freeze, 1979; Bellin et al., 1992). This method involves repeatedly solving the governing equations in a deterministic fashion, varying the stochastic parameters for each run, in order to obtain a set of several realizations for each of the dependent variables. When a sufficient number of realizations is obtained, they can be used to determine the required statistical properties, including the mean system behavior and the standard deviation (Gates and AlZahrani, 1996b). Therefore, the MC simulations require two



models: one which generates realizations for the stochastic parameters, and another (finite-difference model) which deterministically solves the governing flow equations for each realization (Gates and AlZahrani, 1996b). The MC approach is generally accepted as the most robust approach for uncertainty evaluation, as well as the benchmark for comparing other new methods (Scharffenberg and Kavvas, 2011). The full distribution characteristics may be estimated using the MC approach, which is more intuitive than the finite-order methods (Gates and AlZahrani, 1996a). However, the main drawback of the MC approach is its computational expense due to the usual running of a large number of simulations of the process under study in order to obtain accurate results (Dettinger and Wilson, 1981).

To bypass the need for solving the unsteady open-channel flow governing equations several times in order to obtain the statistical descriptions of the dependent variables, a new methodology is proposed in this study in order to solve for the expected system behavior and variability in only one simulation. This methodology involves upscaling the governing stochastic differential equations from the point scale (at which they are originally valid) to the field scale. Ensemble averaging has been a common approach to upscale hydrologic equations that are linear (Gelhar and Axness, 1983; Kitanidis, 1988; Rubin and Dagan, 1989; Kapoor and Gelhar, 1994; Kavvas and Karakas, 1996; Wood and Kavvas, 1999b, a) or nonlinear (Mantoglou and Gelhar, 1987; Tayfur and Kavvas, 1994; Horne and Kavvas, 1997; Dogrul et al., 1998), in which case these equations are averaged to become deterministic differential equations. These developed deterministic differential equations use statistical descriptions, such as the mean and variance, to represent the values of the stochastic parameters (Liang and Kavvas, 2008). However, most of the studies performing the ensemble averaging technique on nonlinear conservation equations used the regular perturbation method, which includes linearization assumptions and which only works for small fluctuations in the dependent variables (Kavvas, 2003). Other techniques that have been applied, which are not limited by small fluctuations, include the decomposition method (Serrano, 1995), a combination of volume averaging with nonlinear dynamics (Duffy, 1996; Duffy and Cusumano, 1998), as well as the theory of fractals and multifractals (Puente, 1996). Nonetheless, due to some limitations of such methods when used for stochastic nonlinear hydrologic processes, the upscaling method used in this study is chosen to be that of Kavvas (2003).

Kavvas (2003) developed general ensemble average conservation equations (to second order) for nonlinear and linear hydrologic processes in order to determine their probabilistic and mean behavior. The "master key" equations developed may be used on any stochastic hydrologic process after being rewritten as one or more linear/nonlinear stochastic ordinary differential equations (ODEs). This utilization leads to a special Lagrangian–Eulerian form of the Fokker–Planck equation (LEFPE) that models the time-space evolution of the probability density of the dependent variables of any nonlinear/linear stochastic dynamic process (Kavvas, 2003). Such a methodology has been successfully applied to many hydrologic processes, including unsaturated water flow (Kim et al., 2005b), root-water uptake (Kim et al., 2005a), solute transport (Liang and Kavvas, 2008), snow accumulation and melt (Ohara et al., 2008), unconfined groundwater flow (Cayar and Kavvas, 2009a, b), as well as kinematic open-channel flow (Ercan and Kavvas, 2012a, b).

Noting that the characteristic forms of the Saint-Venant equations are nonlinear ODEs, it is proposed in this study to apply to them their corresponding master key equation from Kavvas (2003). From this operation the corresponding LEFPE



of the Saint-Venant open-channel flow equations is obtained, thus providing the ability to model the uncertainties of the channel and flow parameters and to compute their effect on the behavior of the system. Therefore, under the appropriate initial and boundary conditions, the probability density functions (PDFs) of the dependent variables can be computed (to exact second order) through the LEFPE, and the ensemble behavior of the system can be described.

The advantages of using the LEFPE in tackling the flood routing problem greatly echo those of the classical Fokker–Planck equation (FPE). In fact, the LEFPE directly solves for the PDFs of the dependent variables of the system in both time and space, it is linear in the variable being solved for (i.e., in the PDF), and unlike the many simulations usually performed for the MC approach, the LEFPE produces the complete ensemble model results with only one single simulation. As such, the LEFPE provides not only the mean and variance of the process, but also a complete description of the evolution

of the dependent variables' PDFs in a computationally efficient manner. Note that the LEFPE does not make any linearization assumptions, it works with a wide-ranging parameter space, and the only assumption about the physical process it makes is the finite correlation time for the process (Ercan and Kavvas, 2012a).

       Therefore, following from the above discussion, the main objective of this study is to apply the upscaling method based on the LEFPE approach in Kavvas (2003) to the characteristic form of the stochastic Saint-Venant equations in order

to derive a new methodology that solves for the probability density of the dependent flow variables, and that quantifies the expected behavior and variability of the system in one shot, instead of running a large number of simulations.

## 2 Saint-Venant equations for unsteady open-channel flow

The Saint-Venant equations, also known as the spatially varied unsteady flow equations (Sturm, 2001), are the two governing equations used to describe an unsteady open-channel flow problem that will be solved using the hydraulic routing

technique (Chow, 1959; Viessman et al., 1977; Sturm, 2001). They consist of the continuity equation and the momentum equation which are used simultaneously in order to solve for the two unknowns (velocity and depth, or discharge and depth). The naming of these equations comes from the French mathematician Adhémar-Jean-Claude Barré de Saint-Venant who published the equations describing one-dimensional unsteady open-channel flow in 1871 (Barré de Saint-Venant, 1871).

       Several assumptions are made when deriving these equations (Viessman et al., 1977; Sturm, 2001), including:

unidirectional flow and uniform cross-sectional velocity, hydrostatic pressure, small channel bed slope, steady state estimation of friction loss, and incompressible flow. Following these assumptions, the Saint-Venant equations for unsteady open-channel flow of an incompressible fluid in a rectangular, prismatic channel (with no lateral inflow/outflow) may be written as follows (Viessman et al., 1977):

*Continuity*     $y\dfrac{\partial V}{\partial x} + V\dfrac{\partial y}{\partial x} + \dfrac{\partial y}{\partial t} = 0$                                         (1)

*Momentum*     $\dfrac{\partial V}{\partial t} + V\dfrac{\partial V}{\partial x} + g\dfrac{\partial y}{\partial x} = g(S_0 - S_f)$                             (2)





where $V$ is the average flow velocity, $y$ is the flow depth, $x$ is the position, $t$ is the time, $S_0$ is the slope of the channel bottom, $S_f$ is the friction slope, and g is the acceleration of gravity. It should be noted that Eqs. (1) and (2) work well for cases which are compatible with the assumptions used during their derivation. When those assumptions are no longer valid, the derived Saint-Venant equations will show some limitations. Such limitations occur, for example, if the flow is not one-dimensional

(e.g., in flood plains or large rivers), if the pressure is non-hydrostatic (e.g., presence of sharp geometric variations/bends or hydraulic jumps), if there are sharp discontinuities (e.g., those caused by weirs or gates), or if there are channel irregularities (Litrico and Fromion, 2009). Therefore, keeping those limitations in mind is crucial for the appropriate implementation of the derived Saint-Venant equations.

## 2.1 Solution methods for the Saint-Venant equations

Since closed-form solutions to the Saint-Venant equations have not been obtained due to the presence of nonlinear terms, it has not been possible to solve these equations analytically except when extreme simplifications are applied (Sturm, 2001; Chaudhry, 2008). As a result, several numerical techniques have been developed in order to solve the Saint-Venant equations deterministically in their full form, without major simplifications.

Finite-difference methods are the most frequently used numerical techniques to solve the unsteady flow equations

(Abbott and Ionescu, 1967; Fread, 1973; Beam and Warming, 1976; Fennema and Chaudhry, 1986; Garcia and Kahawita, 1986; Venutelli, 2002). In such methods, the derivatives of the governing equations are approximated by a finite-difference formulation which is then substituted into the partial differential forms of the equations, thus transforming the governing equations into difference equations that are solved along a fixed rectangular $x$–$t$ grid (Gates and AlZahrani, 1996a). The type of differencing technique determines whether the finite-difference method is explicit or implicit (Szymkiewicz, 2010).

Finite element methods are also available for the solution of unsteady flow equations (Cooley and Moin, 1976; Szymkiewicz, 1991, 1995; Hicks and Steffler, 1995), though they are usually considered to be more effective for two- and three-dimensional flow problems (Szymkiewicz, 1991). In such methods, the domain is divided into small adjacent areas, called finite elements, each of which has a finite number of nodes on its boundaries over which the solution is computed (Szymkiewicz, 2010). While the standard finite element method may not be the most suitable or satisfactory method for

solving unsteady flow problems (Szymkiewicz, 2010), the modified finite element method (Szymkiewicz, 1995) seems to be as effective and robust in solving such problems as the high-order finite-difference methods.

Another approach to solve the Saint-Venant equations may be utilized after realizing that these equations are hyperbolic PDEs (Chaudhry, 2008). This approach is known as the method of characteristics (MOC) (Abbott, 1966), which is one of the earliest and most exact methods for solving hyperbolic PDEs (Tannehill et al., 1997), and which was used early

on for solving the Saint-Venant equations (Amein, 1966; Woolhiser and Liggett, 1967; Lai, 1988). The MOC may be used to transform a hyperbolic PDE into a system of ODEs, which may be simpler to solve (Sturm, 2001). These ODEs are usually divided into two equations: the characteristic equation (i.e., the ODE describing the characteristic path), and the compatibility equation (i.e., the ODE that describes the process behavior along that characteristic path) (Hoffman, 2001).





After this transformation by the MOC, the finite-difference approximations of the derivatives can then be applied to the characteristic form of the hyperbolic PDE, instead of applying them to its original form. Note that in time and one-space dimensions, the characteristic equations represent curves in the *x–t* plane along which information propagates through the solution domain (Hoffman, 2001), and along which discontinuities in the derivatives of the dependent variables propagate (Sturm, 2001).

From the several techniques available to solve for the Saint-Venant equations, the MOC is chosen for this study. This is because, as was mentioned in Sect. 1, the upscaling technique based on the LEFPE approach in Kavvas (2003) can be applied to hydrologic processes that are written as one or more ODEs. As such, it is imperative for the progression of this study to transform the Saint-Venant equations into their characteristic form in order to write them as a system of ODEs so that the LEFPE in Kavvas (2003) can be applied to them. With two characteristic directions, the Saint-Venant equations are transformed by the MOC into a system of four ODEs: two characteristic equations and two corresponding compatibility equations. When finite-difference approximations are applied to the characteristic form of the Saint-Venant equations, the results can be numerically computed along an irregular *x–t* grid formed by the intersection points of the characteristic curves (Gates and AlZahrani, 1996a).

## 2.2 Characteristic form of the Saint-Venant equations

Through a linear combination of the continuity and momentum equations (Eqs. (1) and (2)), the characteristic equations for unsteady open-channel flow of an incompressible fluid in a rectangular, prismatic channel with no lateral inflow can be written as follows (Sturm, 2001):

*Positive characteristic curve ($C_1$)*
$$\frac{dx_1}{dt} = V + c \tag{3}$$

*Flow process condition to be satisfied along $C_1$*
$$\left(\frac{d(V + 2c)}{dt}\right)_1 = g\left(S_{0,1} - S_{f,1}\right) \tag{4}$$

*Negative characteristic curve ($C_2$)*
$$\frac{dx_2}{dt} = V - c \tag{5}$$

*Flow process condition to be satisfied along $C_2$*
$$\left(\frac{d(V - 2c)}{dt}\right)_2 = g\left(S_{0,2} - S_{f,2}\right) \tag{6}$$

where $c$ is the wave celerity which is equal to $\sqrt{gy}$ for a rectangular channel, and $S_{0,2}$ is equal to $S_0(x_2,t)$ (similarly for the other $S$ variables). The remaining variables are defined as in Eqs. (1) and (2).

Equations (3) and (5) represent two different velocity expressions defining the two characteristic directions of the Saint-Venant equations: the former defining the positive characteristic curve ($C_1$) with speed $V + c$, and the latter defining the negative characteristic curve ($C_2$) with speed $V – c$. Equations (4) and (6) represent the compatibility equations for Eqs.



(3) and (5), respectively. Each compatibility equation for the flow process behavior should be satisfied along its corresponding characteristic curve. The subscripts in Eqs. (3) to (6) are used to differentiate between the two total derivative operators which correspond to the two different speeds along $C_1$ and $C_2$. As such, Eqs. (3) to (6) are seen to describe the change of two functions along two different paths: a function $V + 2c$ that varies along $C_1$, and another function $V – 2c$ that

varies along $C_2$. These functions are known as the Riemann invariants (Chaudhry, 2008).

Therefore, the MOC transforms the two governing PDEs into a system of four ODEs that are differentiated with respect to time only, and no longer with respect to space. This transformation provides the ability to use the upscaling technique based on the LEFPE approach in Kavvas (2003) on the Saint-Venant equations in order to derive this study's proposed methodology involving the ensemble-averaged equations of stochastic unsteady open-channel flow.

**3 Ensemble-averaged equations for the stochastic unsteady open-channel flow**

As discussed in Sect. 1, once an uncertainty is introduced through one or more of the parameters of the Saint-Venant equations, the equations have to be solved as stochastic equations in order to model the ensemble behavior and variability of the system. While the MC approach may be used, it is computationally intensive and usually involves a large number of simulations. In this section, a new methodology will be introduced and derived, and a numerical discretization scheme will

be devised for it as well. The proposed methodology aims at obtaining the statistical properties of the dependent variables of the unsteady open-channel flow system in only one simulation. The uncertain parameter chosen for the following derivation is the Manning's roughness coefficient, but similar steps can be followed even when the uncertainty is assumed to arise from other parameters.

**3.1 Development of the Fokker–Planck solution methodology for the Saint-Venant equations**

Following Kavvas (2003), a system of point-scale conservation equations can be written for a dynamical system as follows:

$$\frac{\partial \boldsymbol{H}(\boldsymbol{x},t)}{\partial t} = \boldsymbol{\eta}(\boldsymbol{H}, \mathbf{A}, \boldsymbol{f}; \boldsymbol{x}, t) \qquad (7)$$

where $\boldsymbol{H}(\boldsymbol{x},t)$ is the vector of all state variables of the hydrologic system of equations, $\mathbf{A}(\mathbf{x}, t)$ is the tensor of parameters, $\boldsymbol{f}(\boldsymbol{x}, t)$ is the vector of forcing functions, $\boldsymbol{\eta}$ is a function of $\boldsymbol{H}$, $\mathbf{A}$, and $\boldsymbol{f}$, $\boldsymbol{x}$ is the vector of spatial locations, and $t$ is the time. The initial condition for the above system is given as

$$\boldsymbol{H}(\boldsymbol{x},0) = \boldsymbol{H}_0 \qquad (8)$$

The general Lagrangian–Eulerian form of the Fokker–Planck equation (LEFPE) for the previously defined dynamical system

was developed in Kavvas (2003) to have the following form:



$$\frac{\partial P(\boldsymbol{H}(\boldsymbol{x}_t, t), t)}{\partial t} =$$

$$-\frac{\partial}{\partial H_j}\left\{ P(\boldsymbol{H}(\boldsymbol{x}_t, t), t) \left[ \begin{array}{l} \langle \eta_j(\boldsymbol{H}(\boldsymbol{x}_t, t), \mathbf{A}(\boldsymbol{x}_t, t), \boldsymbol{f}(\boldsymbol{x}_t, t)) \rangle \\ + \int_0^t ds\, \text{Cov}_o\left[ \frac{\partial \eta_j(\boldsymbol{H}(\boldsymbol{x}_t, t), \mathbf{A}(\boldsymbol{x}_t, t), \boldsymbol{f}(\boldsymbol{x}_t, t))}{\partial H_i}; \eta_i(\boldsymbol{H}(\boldsymbol{x}_{t-s}, t-s), \mathbf{A}(\boldsymbol{x}_{t-s}, t-s), \boldsymbol{f}(\boldsymbol{x}_{t-s}, t-s)) \right] \end{array} \right] \right\} \quad (9)$$

$$+\frac{1}{2}\frac{\partial^2}{\partial H_j\,\partial H_i}\left\{ 2P(\boldsymbol{H}(\boldsymbol{x}_t, t), t) \int_0^t ds\, \text{Cov}_o\left[ \eta_j(\boldsymbol{H}(\boldsymbol{x}_t, t), \mathbf{A}(\boldsymbol{x}_t, t), \boldsymbol{f}(\boldsymbol{x}_t, t)); \eta_i(\boldsymbol{H}(\boldsymbol{x}_{t-s}, t-s), \mathbf{A}(\boldsymbol{x}_{t-s}, t-s), \boldsymbol{f}(\boldsymbol{x}_{t-s}, t-s)) \right] \right\}$$

where $P(\boldsymbol{H}(\boldsymbol{x}_t, t), t)$ is the probability density function of the vector of state variables ($\boldsymbol{H}$) at location $\boldsymbol{x}_t$ and at time $t$, the operator $\langle \cdot \rangle$ is the ensemble average operator, $s$ is a time displacement, and $\text{Cov}_o[\cdot]$ is the time-ordered covariance function as shown in the below equation (Van Kampen, 1974):

$$\text{Cov}_o[\eta_j(\boldsymbol{x}, t_1), \eta_i(\boldsymbol{x}, t_2)] = \langle \eta_j(\boldsymbol{x}, t_1)\eta_i(\boldsymbol{x}, t_2) \rangle - \langle \eta_j(\boldsymbol{x}, t_1) \rangle \langle \eta_i(\boldsymbol{x}, t_2) \rangle \quad (10)$$

Note that in Eq. (9), the real space location $\boldsymbol{x}_t$ is known, whereas the Lagrangian location $\boldsymbol{x}_{t-s}$ is unknown. This Lagrangian location can be determined from the known location $\boldsymbol{x}_t$ by using a Lie operator as defined in Kavvas and Karakas (1996):

$$\boldsymbol{x}_{t-s} = \overleftarrow{\exp}\left[ -\int_{t-s}^{t} d\tau\, \langle v_l(\boldsymbol{x}_\tau, \tau) \rangle \frac{\partial}{\partial x_l} \right] \boldsymbol{x}_t \quad (11)$$

where $\overleftarrow{\exp}$ is the time-ordered exponential, and $v_l$ is determined from the characteristic curve equation corresponding to a particular hydrologic/hydraulic conservation equation. In the three-dimensional flow case, $l$ takes on the values 1, 2, and 3. A first-order approximation of Eq. (11) can be written as follows (Kavvas and Karakas, 1996):

$$\boldsymbol{x}_{t-s} = \boldsymbol{x}_t - \int_{t-s}^{t} d\tau\, \langle \boldsymbol{v}(\boldsymbol{x}_\tau, \tau) \rangle \quad (12)$$

where $\boldsymbol{v} = [v_1, v_2, v_3]$ in the general three-dimensional flow case.

Solving the LEFPE, Eq. (9), under the appropriate initial and boundary conditions provides the spatiotemporal evolution of the PDF of the vector of state variables ($\boldsymbol{H}$) for any hydrologic system expressed in terms of Eqs. (7) and (8), thus providing the ensemble behavior and variability of the process with only one simulation. In addition, it is important to note that the LEFPE, a parabolic partial differential equation, is a deterministic equation that is linear in its unknown variable $P(\boldsymbol{H}(\boldsymbol{x}_t, t), t)$, unlike the original hydrologic system which would usually be stochastic and nonlinear. As such, the LEFPE provides great advantages in simplifying the stochastic solution of the hydrologic system being considered.

Since the LEFPE was developed for a system of ODEs (Eq. (7)), and since the characteristic form of the Saint-Venant equations is a system of four nonlinear ODEs (Eqs. (3) to (6)), it is proposed to apply to these equations the



corresponding LEFPE after making some substitutions and adjustments. First, the friction slope ($S_f$) is computed using Manning's formula (Sturm, 2001). Then, the Riemann invariants are defined as follows:

$$V + 2c = \alpha \tag{13}$$

$$V - 2c = \beta \tag{14}$$

As such, Eqs. (3) to (6) can be written as a system of four ODEs in terms of four state variables ($x_1, x_2, \alpha, \beta$):

$$\frac{dx_1}{dt} = \frac{3}{4}\alpha(x_1, t) + \frac{1}{4}\beta(x_1, t) \equiv \eta_{1,t}(x_1, t) = \eta_{1,t} \tag{15}$$

$$\left(\frac{d\alpha}{dt}\right)_1 = g\,S_0(x_1, t) - \frac{g\,n^2(x_1, t)}{k^2} \cdot \frac{\frac{1}{4}[\alpha(x_1, t) + \beta(x_1, t)]^2}{R^{4/3}(\alpha(x_1, t), \beta(x_1, t), b; x_1, t)} \equiv \eta_{\alpha,t}(x_1, t) = \eta_{\alpha,t} \tag{16}$$

$$\frac{dx_2}{dt} = \frac{1}{4}\alpha(x_2, t) + \frac{3}{4}\beta(x_2, t) \equiv \eta_{2,t}(x_2, t) = \eta_{2,t} \tag{17}$$

$$\left(\frac{d\beta}{dt}\right)_2 = g\,S_0(x_2, t) - \frac{g\,n^2(x_2, t)}{k^2} \cdot \frac{\frac{1}{4}[\alpha(x_2, t) + \beta(x_2, t)]^2}{R^{4/3}(\alpha(x_2, t), \beta(x_2, t), b; x_2, t)} \equiv \eta_{\beta,t}(x_2, t) = \eta_{\beta,t} \tag{18}$$

where $R$ denotes the hydraulic radius, $n$ denotes Manning's roughness coefficient, and k denotes the conversion factor

between SI and US units for Manning's formula. Note that the width of the channel ($b$) is not a function of $x$ or $t$ since the equations are derived for a prismatic channel. Moreover, the gravitational acceleration (g) and the conversion coefficient of Manning's equation (k) are also not functions of $x$ or $t$ since they are constants. Equations (15) to (18) are now in the form of Eq. (7), with an $\eta$ function defined for each of the four ODEs, as shown on their right-hand side. As such, it is clear that for a vector of state variables $\boldsymbol{H} = [x_1, x_2, \alpha, \beta]$, there is a vector of functions $\boldsymbol{\eta} = [\eta_{1,t}, \eta_{2,t}, \eta_{\alpha,t}, \eta_{\beta,t}]$. The subscripts for the $\eta$

functions shown in Eqs. (15) to (18) represent the following: subscript 1 represents the positive characteristic direction ($C_1$), subscript 2 represents the negative characteristic direction ($C_2$), subscript $\alpha$ represents the compatibility equation along $C_1$, and subscript $\beta$ represents the compatibility equation along $C_2$.

Note that in equations to follow, the $\eta$ functions will be interchangeably represented by one of their two forms shown on the right-hand side of Eqs. (15) to (18), whereas $P(x_1, x_2, \alpha, \beta, t)$ may be substituted by $P$ for simplicity. Such

typographical simplifications will be used in order to reduce the used space and to increase the readability and simplicity of large equations. Therefore, considering some of the above typographical simplifications, and applying the general LEFPE in Eq. (9) to Eqs. (15) to (18), the LEFPE for the Saint-Venant equations that would solve for the multivariate PDF of the hydrologic state vector can be written as shown in Eq. (19) below.



$$\frac{\partial P(x_1, x_2, \alpha, \beta, t)}{\partial t} =$$

$$-\frac{\partial}{\partial x_1}\left\{ P(x_1,x_2,\alpha,\beta,t) \begin{bmatrix} \langle \eta_{1,t}(x_1,t)\rangle \\ +\int_0^t ds\, \text{Cov}_o\left[\frac{\partial \eta_{1,t}(x_1,t)}{\partial x_1};\eta_{1,t-s}(x_1,t-s)\right] + \int_0^t ds\, \text{Cov}_o\left[\frac{\partial \eta_{1,t}(x_1,t)}{\partial x_2};\eta_{2,t-s}(x_2,t-s)\right] \\ +\int_0^t ds\, \text{Cov}_o\left[\frac{\partial \eta_{1,t}(x_1,t)}{\partial \alpha};\eta_{\alpha,t-s}(x_1,t-s)\right] + \int_0^t ds\, \text{Cov}_o\left[\frac{\partial \eta_{1,t}(x_1,t)}{\partial \beta};\eta_{\beta,t-s}(x_2,t-s)\right] \end{bmatrix} \right\}$$

$$-\frac{\partial}{\partial x_2}\left\{ P(x_1,x_2,\alpha,\beta,t) \begin{bmatrix} \langle \eta_{2,t}(x_2,t)\rangle \\ +\int_0^t ds\, \text{Cov}_o\left[\frac{\partial \eta_{2,t}(x_2,t)}{\partial x_2};\eta_{2,t-s}(x_2,t-s)\right] + \int_0^t ds\, \text{Cov}_o\left[\frac{\partial \eta_{2,t}(x_2,t)}{\partial x_1};\eta_{1,t-s}(x_1,t-s)\right] \\ +\int_0^t ds\, \text{Cov}_o\left[\frac{\partial \eta_{2,t}(x_2,t)}{\partial \alpha};\eta_{\alpha,t-s}(x_1,t-s)\right] + \int_0^t ds\, \text{Cov}_o\left[\frac{\partial \eta_{2,t}(x_2,t)}{\partial \beta};\eta_{\beta,t-s}(x_2,t-s)\right] \end{bmatrix} \right\}$$

$$-\frac{\partial}{\partial \alpha}\left\{ P(x_1,x_2,\alpha,\beta,t) \begin{bmatrix} \langle \eta_{\alpha,t}(x_1,t)\rangle \\ +\int_0^t ds\, \text{Cov}_o\left[\frac{\partial \eta_{\alpha,t}(x_1,t)}{\partial \alpha};\eta_{\alpha,t-s}(x_1,t-s)\right] + \int_0^t ds\, \text{Cov}_o\left[\frac{\partial \eta_{\alpha,t}(x_1,t)}{\partial x_1};\eta_{1,t-s}(x_1,t-s)\right] \\ +\int_0^t ds\, \text{Cov}_o\left[\frac{\partial \eta_{\alpha,t}(x_1,t)}{\partial x_2};\eta_{2,t-s}(x_2,t-s)\right] + \int_0^t ds\, \text{Cov}_o\left[\frac{\partial \eta_{\alpha,t}(x_1,t)}{\partial \beta};\eta_{\beta,t-s}(x_2,t-s)\right] \end{bmatrix} \right\}$$

$$-\frac{\partial}{\partial \beta}\left\{ P(x_1,x_2,\alpha,\beta,t) \begin{bmatrix} \langle \eta_{\beta,t}(x_2,t)\rangle \\ +\int_0^t ds\, \text{Cov}_o\left[\frac{\partial \eta_{\beta,t}(x_2,t)}{\partial \beta};\eta_{\beta,t-s}(x_2,t-s)\right] + \int_0^t ds\, \text{Cov}_o\left[\frac{\partial \eta_{\beta,t}(x_2,t)}{\partial x_1};\eta_{1,t-s}(x_{1,t-s},t-s)\right] \\ +\int_0^t ds\, \text{Cov}_o\left[\frac{\partial \eta_{\beta,t}(x_2,t)}{\partial x_2};\eta_{2,t-s}(x_2,t-s)\right] + \int_0^t ds\, \text{Cov}_o\left[\frac{\partial \eta_{\beta,t}(x_2,t)}{\partial \alpha};\eta_{\alpha,t-s}(x_{1,t-s},t-s)\right] \end{bmatrix} \right\}$$

$$+\frac{1}{2}\frac{\partial^2}{\partial x_1^2}\left\{ 2P(x_1,x_2,\alpha,\beta,t)\int_0^t ds\, \text{Cov}_o[\eta_{1,t}(x_1,t);\eta_{1,t-s}(x_1,t-s)] \right\}$$

$$+\frac{1}{2}\frac{\partial^2}{\partial x_2^2}\left\{ 2P(x_1,x_2,\alpha,\beta,t)\int_0^t ds\, \text{Cov}_o[\eta_{2,t}(x_2,t);\eta_{2,t-s}(x_2,t-s)] \right\}$$

$$+\frac{1}{2}\frac{\partial^2}{\partial \alpha^2}\left\{ 2P(x_1,x_2,\alpha,\beta,t)\int_0^t ds\, \text{Cov}_o[\eta_{\alpha,t}(x_1,t);\eta_{\alpha,t-s}(x_1,t-s)] \right\}$$

$$+\frac{1}{2}\frac{\partial^2}{\partial \beta^2}\left\{ 2P(x_1,x_2,\alpha,\beta,t)\int_0^t ds\, \text{Cov}_o[\eta_{\beta,t}(x_2,t);\eta_{\beta,t-s}(x_2,t-s)] \right\}$$

$+ \text{other cross} - \text{covariance dispersion terms}$

(19)





Note that the LEFPE has the form of an advection-diffusion equation. In Eq. (19), the first four terms on the right-hand side represent the advection terms, while the remaining terms represent the diffusion terms. Within the advection terms, the expected values of the $\eta$ functions are the mean convection coefficients, while the integrals of the ordered covariance functions added to them are the convection correction terms. However, it was shown in a study by Kavvas and Wu (2002), which used a similar approach but was applied to solute transport, that the convection correction terms are negligible when compared to the mean convection term. As a result, the magnitudes of the expectations of the $\eta$ functions are much larger than those of the integral terms in the advection portion of the LEFPE, thus allowing the removal of these integral terms. As an example, this simplification can be mathematically represented for the first advection term as shown in Eq. (20) below, and is similarly applied to the other advection terms.

$$
\langle \eta_{1,t} \rangle \gg Cov_o \left[ \frac{\partial \eta_{1,t}}{\partial x_1} ; \eta_{1,t-s} \right]; \qquad \langle \eta_{1,t} \rangle \gg Cov_o \left[ \frac{\partial \eta_{1,t}}{\partial x_2} ; \eta_{2,t-s} \right]
$$
$$
\langle \eta_{1,t} \rangle \gg Cov_o \left[ \frac{\partial \eta_{1,t}}{\partial \alpha} ; \eta_{\alpha,t-s} \right]; \qquad \langle \eta_{1,t} \rangle \gg Cov_o \left[ \frac{\partial \eta_{1,t}}{\partial \beta} ; \eta_{\beta,t-s} \right]
$$
(20)

Moreover, note that the cross-covariance dispersion terms have not been explicitly written in Eq. (19). These terms involve the ordered covariance between two different $\eta$ functions, $\eta_i$ and $\eta_j$, where $i \neq j$. Examples of those include:

$$
Cov_o \left[ \eta_{1,t}(x_1, t) ; \eta_{2,t-s}(x_2, t-s) \right]; \qquad Cov_o \left[ \eta_{1,t}(x_1, t) ; \eta_{\beta,t-s}(x_2, t-s) \right]
$$
(21)

However, all of the four $\eta$ functions ($\eta_{1,t}$ ; $\eta_{2,t}$ ; $\eta_{\alpha,t}$ ; $\eta_{\beta,t}$) are functions of the state stochastic variables ($x_1$; $x_2$; $\alpha$; $\beta$). It has been shown in Liang and Kavvas (2008) that the covariance between any two of the different $\eta$ functions is substantially smaller in magnitude when compared to the autocovariance of the $\eta$ function of any one state variable. This leads to another simplification in which all cross-covariance terms, similar to those in Eq. (21), are neglected and removed from the main equation.

A final simplification to the LEFPE concerns the covariance terms in the diffusion coefficients. This involves the assumption that the $\eta_i$ random functions have short memory with respect to $t$, and thus may be approximated as delta-correlated. As a result, the covariance integral term for any of the four functions can be written as follows (where $\delta(s)$ is the Dirac delta function):

$$
\int_0^t ds \, Cov_o \left[ \eta_{i,t} ; \eta_{i,t-s} \right] = \int_0^t Cov_o \left[ \eta_{i,t} ; \eta_{i,t-s} \right] \delta(s) \, ds = Var \left[ \eta_{i,t} \right]
$$
(22)

Note that under the assumption of a delta-correlated covariance, the nonlocal LEFPE reduces to the classical FPE (as it will be called henceforth), which is simpler to apply. Including all of the simplifications discussed above, Eq. (19) can be written as shown in Eq. (23).






$$\frac{\partial P(x_1, x_2, \alpha, \beta, t)}{\partial t} = -\frac{\partial}{\partial x_1}\{P \langle \eta_{1,t}(x_1, t)\rangle\} + \frac{1}{2}\frac{\partial^2}{\partial x_1^2}\{2P \ \text{Var}[\eta_{1,t}(x_1, t)]\}$$

$$-\frac{\partial}{\partial x_2}\{P \langle \eta_{2,t}(x_2, t)\rangle\} + \frac{1}{2}\frac{\partial^2}{\partial x_2^2}\{2P \ \text{Var}[\eta_{2,t}(x_2, t)]\}$$

$$-\frac{\partial}{\partial \alpha}\{P \langle \eta_{\alpha,t}(x_1, t)\rangle\} + \frac{1}{2}\frac{\partial^2}{\partial \alpha^2}\{2P \ \text{Var}[\eta_{\alpha,t}(x_1, t)]\}$$

$$-\frac{\partial}{\partial \beta}\{P \langle \eta_{\beta,t}(x_2, t)\rangle\} + \frac{1}{2}\frac{\partial^2}{\partial \beta^2}\{2P \ \text{Var}[\eta_{\beta,t}(x_2, t)]\}$$

(23)

The validity of the preceding assumptions will be checked when the results of the proposed FPE methodology are compared against the corresponding results obtained from the MC approach. With the final version of the FPE being ready, the last step is to determine the detailed expressions of the expectations and variances of the $\eta$ functions in Eq. (23). After expanding these expressions based on their equivalence as denoted in Eqs. (15) to (18), and after some manipulation, Eq.

5   (23) may be written as follows:

$$\frac{\partial P(x_1, x_2, \alpha, \beta, t)}{\partial t} =$$

$$-\frac{\partial}{\partial x_1}\left\{P\left[\frac{3}{4}\langle\alpha(x_1, t)\rangle + \frac{1}{4}\langle\beta(x_1, t)\rangle\right]\right\}$$

$$-\frac{\partial}{\partial x_2}\left\{P\left[\frac{1}{4}\langle\alpha(x_2, t)\rangle + \frac{3}{4}\langle\beta(x_2, t)\rangle\right]\right\}$$

$$-\frac{\partial}{\partial \alpha}\left\{P\left[g\,S_0 - \frac{g}{4k^2}\left(\frac{2}{b}\right)^{4/3}\left\langle n^2(x_1, t)\cdot[\alpha(x_1, t) + \beta(x_1, t)]^2\cdot\left\{\frac{8gb}{[\alpha(x_1, t) - \beta(x_1, t)]^2} + 1\right\}^{4/3}\right\rangle\right]\right\}$$

$$-\frac{\partial}{\partial \beta}\left\{P\left[g\,S_0 - \frac{g}{4k^2}\left(\frac{2}{b}\right)^{4/3}\left\langle n^2(x_2, t)\cdot[\alpha(x_2, t) + \beta(x_2, t)]^2\cdot\left\{\frac{8gb}{[\alpha(x_2, t) - \beta(x_2, t)]^2} + 1\right\}^{4/3}\right\rangle\right]\right\}$$

$$+\frac{\partial^2}{\partial x_1^2}\left\{P\left[\left(\frac{9}{16}\right)\text{Var}[\alpha(x_1, t)] + \left(\frac{1}{16}\right)\text{Var}[\beta(x_1, t)] + \left(\frac{3}{8}\right)\text{Cov}[\alpha(x_1, t), \beta(x_1, t)]\right]\right\}$$

$$+\frac{\partial^2}{\partial x_2^2}\left\{P\left[\left(\frac{1}{16}\right)\text{Var}[\alpha(x_2, t)] + \left(\frac{9}{16}\right)\text{Var}[\beta(x_2, t)] + \left(\frac{3}{8}\right)\text{Cov}[\alpha(x_2, t), \beta(x_2, t)]\right]\right\}$$

$$+\frac{\partial^2}{\partial \alpha^2}\left\{P\left[\frac{g^2}{16k^4}\left(\frac{2}{b}\right)^{8/3}\text{Var}\left[n^2(x_1, t)\cdot[\alpha(x_1, t) + \beta(x_1, t)]^2\cdot\left\{\frac{8gb}{[\alpha(x_1, t) - \beta(x_1, t)]^2} + 1\right\}^{4/3}\right]\right]\right\}$$

$$+\frac{\partial^2}{\partial \beta^2}\left\{P\left[\frac{g^2}{16k^4}\left(\frac{2}{b}\right)^{8/3}\text{Var}\left[n^2(x_2, t)\cdot[\alpha(x_2, t) + \beta(x_2, t)]^2\cdot\left\{\frac{8gb}{[\alpha(x_2, t) - \beta(x_2, t)]^2} + 1\right\}^{4/3}\right]\right]\right\}$$

(24)

Denoting the advection terms with *F* and the diffusion terms with *D*, Eq. (24) can be written in a simplified form as follows:

$$\frac{\partial P(x_1, x_2, \alpha, \beta, t)}{\partial t} = -\frac{\partial}{\partial x_1}F_1 P - \frac{\partial}{\partial x_2}F_2 P - \frac{\partial}{\partial \alpha}F_\alpha P - \frac{\partial}{\partial \beta}F_\beta P$$

$$+\frac{\partial^2}{\partial x_1^2}D_1 P + \frac{\partial^2}{\partial x_2^2}D_2 P + \frac{\partial^2}{\partial \alpha^2}D_\alpha P + \frac{\partial^2}{\partial \beta^2}D_\beta P$$

(25)



Equation (25) is the final analytical form of the FPE methodology proposed in this study for the probabilistic solution of the stochastic Saint-Venant equations in one simulation. The advection-diffusion form of Eq. (25) is clear, in which the $F$ terms are the advection (drift) coefficients, and the $D$ terms are their corresponding diffusion coefficients. With the mathematical equations for the FPE methodology being derived, the next step is to find a numerical scheme with which Eq. (25) may be computed.

### 3.2 Numerical solution for the proposed Fokker–Planck equation methodology

In order to apply the derived FPE methodology, the FPE represented in Eq. (25) must be solved using an appropriate numerical scheme. Chang and Cooper (1970) developed a practical numerical differencing scheme for the solution of the one-dimensional classical FPE. It is a widely used scheme that ensures the non-negativity of the solution, the conservation of the particles (in the absence of any external sources or sinks), and the exact representation of the analytical solution upon equilibration. As a result, the Chang–Cooper scheme is highly accurate with a relatively small number of required mesh nodes. While Chang and Cooper (1970) developed and applied that scheme to a one-dimensional FPE, Kim et al. (2005a) generalized and applied the Chang–Cooper scheme to a two-dimensional FPE. In a similar manner, for the case of this study, an attempt is made to generalize the Chang–Cooper scheme to the four-dimensional FPE shown in Eq. (25). Note that the finite-difference method is used for the discretization.

First, Eq. (25) is rewritten as follows:

$$\frac{\partial P(x_1, x_2, \alpha, \beta, t)}{\partial t} = -\frac{\partial}{\partial x_1}\left[F_1 P - D_1 \frac{\partial}{\partial x_1}P\right] - \frac{\partial}{\partial x_2}\left[F_2 P - D_2 \frac{\partial}{\partial x_2}P\right] - \frac{\partial}{\partial \alpha}\left[F_\alpha P - D_\alpha \frac{\partial}{\partial \alpha}P\right] - \frac{\partial}{\partial \beta}\left[F_\beta P - D_\beta \frac{\partial}{\partial \beta}P\right] \quad (26)$$

$$\frac{\partial P(x_1, x_2, \alpha, \beta, t)}{\partial t} = -\frac{\partial}{\partial x_1}J_1 - \frac{\partial}{\partial x_2}J_2 - \frac{\partial}{\partial \alpha}J_\alpha - \frac{\partial}{\partial \beta}J_\beta \quad (27)$$

Equation (27) is in the form of the continuity equation, in which the $J$ parameters may be interpreted as the probability flux or probability current, whereas $P$ (i.e., the probability density function) is considered as the state variable. This equation can then be discretized in the following implicit manner:

$$\frac{P_{i,j,k,l}^{n+1} - P_{i,j,k,l}^{n}}{\Delta t} = -\frac{J_{1;\,i+\frac{1}{2},j,k,l}^{n+1} - J_{1;\,i-\frac{1}{2},j,k,l}^{n+1}}{\Delta x_1} - \frac{J_{2;\,i,j+\frac{1}{2},k,l}^{n+1} - J_{2;\,i,j-\frac{1}{2},k,l}^{n+1}}{\Delta x_2} - \frac{J_{\alpha;\,i,j,k+\frac{1}{2},l}^{n+1} - J_{\alpha;\,i,j,k-\frac{1}{2},l}^{n+1}}{\Delta \alpha} - \frac{J_{\beta;\,i,j,k,l+\frac{1}{2}}^{n+1} - J_{\beta;\,i,j,k,l-\frac{1}{2}}^{n+1}}{\Delta \beta} \quad (28)$$

where $i$: 0, 1, 2, …, $N_I$ denotes the domain of $x_1$ in the direction of the $C_1$ curve; $j$: 0, 1, 2, …, $N_J$ denotes the domain of $x_2$ in the direction of the $C_2$ curve; $k$: 0, 1, 2, …, $N_K$ denotes the domain of $\alpha$; $l$: 0, 1, 2, …, $N_L$ denotes the domain of $\beta$; and $n$: 0, 1, 2, … denotes the domain of time $t$. Following the Chang–Cooper scheme, the expressions for computing variables between two nodes (e.g., at $i + 1/2$ or at $k + 1/2$) are defined by the following:

$$P_{i+\frac{1}{2},j,k,l}^{n+1} = \left(1 - \delta_{1;\,i}^{n+1}\right)P_{i+1,j,k,l}^{n+1} + \delta_{1;\,i}^{n+1}P_{i,j,k,l}^{n+1} \quad (29)$$





$$P^{n+1}_{i,j+\frac{1}{2},k,l} = \left(1 - \delta^{n+1}_{2;\,j}\right)P^{n+1}_{i,j+1,k,l} + \delta^{n+1}_{2;\,j}P^{n+1}_{i,j,k,l} \tag{30}$$

$$P^{n+1}_{i,j,k+\frac{1}{2},l} = \left(1 - \delta^{n+1}_{\alpha;\,k}\right)P^{n+1}_{i,j,k+1,l} + \delta^{n+1}_{\alpha;\,k}P^{n+1}_{i,j,k,l} \tag{31}$$

$$P^{n+1}_{i,j,k,l+\frac{1}{2}} = \left(1 - \delta^{n+1}_{\beta;\,l}\right)P^{n+1}_{i,j,k,l+1} + \delta^{n+1}_{\beta;\,l}P^{n+1}_{i,j,k,l} \tag{32}$$

where the $\delta$ values are weighting factors. In the one-dimensional case, Chang and Cooper (1970) developed the expressions of these weighting factors in a manner that would ensure the non-negativity of the PDF solution and that would give proper equilibration. In a similar manner, the same steps are followed for this study in order to derive the expressions for the four $\delta$ values present in Eqs. (29) to (32). These expressions are shown below:

$$\delta^{n+1}_{1;\,i} = \frac{D^n_{1;\,i+\frac{1}{2},j,k,l} - \left(D^n_{1;\,i+\frac{1}{2},j,k,l} - \Delta x_1 F^n_{1;\,i+\frac{1}{2},j,k,l}\right)\exp\left[\Delta x_1 \frac{F^n_{1;\,i+\frac{1}{2},j,k,l}}{D^n_{1;\,i+\frac{1}{2},j,k,l}}\right]}{\Delta x_1 F^n_{1;\,i+\frac{1}{2},j,k,l}\left\{\exp\left[\Delta x_1 \frac{F^n_{1;\,i+\frac{1}{2},j,k,l}}{D^n_{1;\,i+\frac{1}{2},j,k,l}}\right] - 1\right\}} \tag{33}$$

$$\delta^{n+1}_{2;\,j} = \frac{D^n_{2;\,i,j+\frac{1}{2},k,l} - \left(D^n_{2;\,i,j+\frac{1}{2},k,l} - \Delta x_2 F^n_{2;\,i,j+\frac{1}{2},k,l}\right)\exp\left[\Delta x_2 \frac{F^n_{2;\,i,j+\frac{1}{2},k,l}}{D^n_{2;\,i,j+\frac{1}{2},k,l}}\right]}{\Delta x_2 F^n_{2;\,i,j+\frac{1}{2},k,l}\left\{\exp\left[\Delta x_2 \frac{F^n_{2;\,i,j+\frac{1}{2},k,l}}{D^n_{2;\,i,j+\frac{1}{2},k,l}}\right] - 1\right\}} \tag{34}$$

$$\delta^{n+1}_{\alpha;\,k} = \frac{D^n_{\alpha;\,i,j,k+\frac{1}{2},l} - \left(D^n_{\alpha;\,i,j,k+\frac{1}{2},l} - \Delta\alpha F^n_{\alpha;\,i,j,k+\frac{1}{2},l}\right)\exp\left[\Delta\alpha \frac{F^n_{\alpha;\,i,j,k+\frac{1}{2},l}}{D^n_{\alpha;\,i,j,k+\frac{1}{2},l}}\right]}{\Delta\alpha F^n_{\alpha;\,i,j,k+\frac{1}{2},l}\left\{\exp\left[\Delta\alpha \frac{F^n_{\alpha;\,i,j,k+\frac{1}{2},l}}{D^n_{\alpha;\,i,j,k+\frac{1}{2},l}}\right] - 1\right\}} \tag{35}$$

$$\delta^{n+1}_{\beta;\,l} = \frac{D^n_{\beta;\,i,j,k,l+\frac{1}{2}} - \left(D^n_{\beta;\,i,j,k,l+\frac{1}{2}} - \Delta\beta F^n_{\beta;\,i,j,k,l+\frac{1}{2}}\right)\exp\left[\Delta\beta \frac{F^n_{\beta;\,i,j,k,l+\frac{1}{2}}}{D^n_{\beta;\,i,j,k,l+\frac{1}{2}}}\right]}{\Delta\beta F^n_{\beta;\,i,j,k,l+\frac{1}{2}}\left\{\exp\left[\Delta\beta \frac{F^n_{\beta;\,i,j,k,l+\frac{1}{2}}}{D^n_{\beta;\,i,j,k,l+\frac{1}{2}}}\right] - 1\right\}} \tag{36}$$




Moreover, following the Chang–Cooper scheme, the expressions for the *J* parameters may be derived to be represented as follows:

$$
J_{1;\,i+\frac{1}{2},j,k,l}^{n+1} = \left[ F_{1;\,i+\frac{1}{2},j,k,l}^{n}\left(1-\delta_{1;\,i}^{n+1}\right) - \frac{D_{1;\,i+\frac{1}{2},j,k,l}^{n}}{\Delta x_1} \right] P_{i+1,j,k,l}^{n+1} + \left[ F_{1;\,i+\frac{1}{2},j,k,l}^{n}\delta_{1;\,i}^{n+1} + \frac{D_{1;\,i+\frac{1}{2},j,k,l}^{n}}{\Delta x_1} \right] P_{i,j,k,l}^{n+1}
\tag{37}
$$

$$
J_{2;\,i,j+\frac{1}{2},k,l}^{n+1} = \left[ F_{2;\,i,j+\frac{1}{2},k,l}^{n}\left(1-\delta_{2;\,j}^{n+1}\right) - \frac{D_{2;\,i,j+\frac{1}{2},k,l}^{n}}{\Delta x_2} \right] P_{i,j+1,k,l}^{n+1} + \left[ F_{2;\,i,j+\frac{1}{2},k,l}^{n}\delta_{2;\,j}^{n+1} + \frac{D_{2;\,i,j+\frac{1}{2},k,l}^{n}}{\Delta x_2} \right] P_{i,j,k,l}^{n+1}
\tag{38}
$$

$$
J_{\alpha;\,i,j,k+\frac{1}{2},l}^{n+1} = \left[ F_{\alpha;\,i,j,k+\frac{1}{2},l}^{n}\left(1-\delta_{\alpha;\,k}^{n+1}\right) - \frac{D_{\alpha;\,i,j,k+\frac{1}{2},l}^{n}}{\Delta \alpha} \right] P_{i,j,k+1,l}^{n+1} + \left[ F_{\alpha;\,i,j,k+\frac{1}{2},l}^{n}\delta_{\alpha;\,k}^{n+1} + \frac{D_{\alpha;\,i,j,k+\frac{1}{2},l}^{n}}{\Delta \alpha} \right] P_{i,j,k,l}^{n+1}
\tag{39}
$$

$$
J_{\beta;\,i,j,k,l+\frac{1}{2}}^{n+1} = \left[ F_{\beta;\,i,j,k,l+\frac{1}{2}}^{n}\left(1-\delta_{\beta;\,l}^{n+1}\right) - \frac{D_{\beta;\,i,j,k,l+\frac{1}{2}}^{n}}{\Delta \beta} \right] P_{i,j,k,l+1}^{n+1} + \left[ F_{\beta;\,i,j,k,l+\frac{1}{2}}^{n}\delta_{\beta;\,l}^{n+1} + \frac{D_{\beta;\,i,j,k,l+\frac{1}{2}}^{n}}{\Delta \beta} \right] P_{i,j,k,l}^{n+1}
\tag{40}
$$

Equations (37) to (40) can then be substituted into the discretized FPE, Eq. (28), in order to provide the implicit finite-difference form of the FPE methodology, shown in Eq. (41) below, which can be numerically solved. Note that in the expression of Eq. (41), each subscript (*i*, *j*, *k*, *l*) that is not followed by a +1/2 or –1/2 is dropped from the expressions of *F* and *D* for readability purposes; e.g., $F_{2;\,i,j+\frac{1}{2},k,l}^{n}$ is simplified and written as $F_{2;\,j+\frac{1}{2}}^{n}$.





$$
\begin{aligned}
P_{i,j,k,l}^{n} = {} & \left\{
\begin{aligned}
& 1 + \frac{\Delta t}{\Delta x_1}\delta_{1;\,i}^{n+1}F_{1;\,i+\frac{1}{2}}^{n} + \frac{\Delta t}{(\Delta x_1)^2}D_{1;\,i+\frac{1}{2}}^{n} - \frac{\Delta t}{\Delta x_1}\left(1-\delta_{1;\,i-1}^{n+1}\right)F_{1;\,i-\frac{1}{2}}^{n} + \frac{\Delta t}{(\Delta x_1)^2}D_{1;\,i-\frac{1}{2}}^{n} \\
& + \frac{\Delta t}{\Delta x_2}\delta_{2;\,j}^{n+1}F_{2;\,j+\frac{1}{2}}^{n} + \frac{\Delta t}{(\Delta x_2)^2}D_{2;\,j+\frac{1}{2}}^{n} - \frac{\Delta t}{\Delta x_2}\left(1-\delta_{2;\,j-1}^{n+1}\right)F_{2;\,j-\frac{1}{2}}^{n} + \frac{\Delta t}{(\Delta x_2)^2}D_{2;\,j-\frac{1}{2}}^{n} \\
& + \frac{\Delta t}{\Delta \alpha}\delta_{\alpha;\,k}^{n+1}F_{\alpha;\,k+\frac{1}{2}}^{n} + \frac{\Delta t}{(\Delta \alpha)^2}D_{\alpha;\,k+\frac{1}{2}}^{n} - \frac{\Delta t}{\Delta \alpha}\left(1-\delta_{\alpha;\,k-1}^{n+1}\right)F_{\alpha;\,k-\frac{1}{2}}^{n} + \frac{\Delta t}{(\Delta \alpha)^2}D_{\alpha;\,k-\frac{1}{2}}^{n} \\
& + \frac{\Delta t}{\Delta \beta}\delta_{\beta;\,l}^{n+1}F_{\beta;\,l+\frac{1}{2}}^{n} + \frac{\Delta t}{(\Delta \beta)^2}D_{\beta;\,l+\frac{1}{2}}^{n} - \frac{\Delta t}{\Delta \beta}\left(1-\delta_{\beta;\,l-1}^{n+1}\right)F_{\beta;\,l-\frac{1}{2}}^{n} + \frac{\Delta t}{(\Delta \beta)^2}D_{\beta;\,l-\frac{1}{2}}^{n}
\end{aligned}
\right\} P_{i,j,k,l}^{n+1} \\
& + \left[\frac{\Delta t}{\Delta x_1}\left(1-\delta_{1;\,i}^{n+1}\right)F_{1;\,i+\frac{1}{2}}^{n} - \frac{\Delta t}{(\Delta x_1)^2}D_{1;\,i+\frac{1}{2}}^{n}\right]P_{i+1,j,k,l}^{n+1} \\
& + \left[\frac{\Delta t}{\Delta x_2}\left(1-\delta_{2;\,j}^{n+1}\right)F_{2;\,j+\frac{1}{2}}^{n} - \frac{\Delta t}{(\Delta x_2)^2}D_{2;\,j+\frac{1}{2}}^{n}\right]P_{i,j+1,k,l}^{n+1} \\
& + \left[\frac{\Delta t}{\Delta \alpha}\left(1-\delta_{\alpha;\,k}^{n+1}\right)F_{\alpha;\,k+\frac{1}{2}}^{n} - \frac{\Delta t}{(\Delta \alpha)^2}D_{\alpha;\,k+\frac{1}{2}}^{n}\right]P_{i,j,k+1,l}^{n+1} \\
& + \left[\frac{\Delta t}{\Delta \beta}\left(1-\delta_{\beta;\,l}^{n+1}\right)F_{\beta;\,l+\frac{1}{2}}^{n} - \frac{\Delta t}{(\Delta \beta)^2}D_{\beta;\,l+\frac{1}{2}}^{n}\right]P_{i,j,k,l+1}^{n+1} \\
& + \left[-\frac{\Delta t}{\Delta x_1}\delta_{1;\,i-1}^{n+1}F_{1;\,i-\frac{1}{2}}^{n} - \frac{\Delta t}{(\Delta x_1)^2}D_{1;\,i-\frac{1}{2}}^{n}\right]P_{i-1,j,k,l}^{n+1} \\
& + \left[-\frac{\Delta t}{\Delta x_2}\delta_{2;\,j-1}^{n+1}F_{2;\,j-\frac{1}{2}}^{n} - \frac{\Delta t}{(\Delta x_2)^2}D_{2;\,j-\frac{1}{2}}^{n}\right]P_{i,j-1,k,l}^{n+1} \\
& + \left[-\frac{\Delta t}{\Delta \alpha}\delta_{\alpha;\,k-1}^{n+1}F_{\alpha;\,k-\frac{1}{2}}^{n} - \frac{\Delta t}{(\Delta \alpha)^2}D_{\alpha;\,k-\frac{1}{2}}^{n}\right]P_{i,j,k-1,l}^{n+1} \\
& + \left[-\frac{\Delta t}{\Delta \beta}\delta_{\beta;\,l-1}^{n+1}F_{\beta;\,l-\frac{1}{2}}^{n} - \frac{\Delta t}{(\Delta \beta)^2}D_{\beta;\,l-\frac{1}{2}}^{n}\right]P_{i,j,k,l-1}^{n+1}
\end{aligned}
\tag{41}
$$

Note that the derived FPE discretized in Eq. (41) was originally described and represented using the characteristic method. As a result, the computed values of the state variables at a new time step would be solved at the intersection of the characteristic curves $C_1$ and $C_2$ (see Eqs. (3) and (5)). In a similar manner, the values of $P$ to be solved for in Eq. (41) should be those corresponding to the positions of intersection between $C_1$ and $C_2$, i.e., when $x_1 = x_2$. Hence, additional simplifications can be applied to Eq. (41), including $x_1 = x_2 = x$, and $\Delta x_1 = \Delta x_2 = \Delta x$. As such, since the variables $x_1$ and $x_2$ are now represented by one variable $x$, which is the intersection position, their corresponding $i$ and $j$ subscript representations can be merged into a single representation, $h$, thus reducing the equation from four to three dimensions $(x, \alpha, \beta)$. Therefore, the PDF $P_{i,j,k,l}$ can now be represented as $P_{h,k,l}$, where $h$ represents the domain of the intersection position $x$. However, note that this does not affect the computations of the $F$, $D$, and $\delta$ parameters for $x_1$ and $x_2$ since each one has its own different expression for its calculation. With the above changes, Eq. (41) can finally be rewritten as shown in Eq. (42).



$$
\begin{aligned}
P_{h,k,l}^{n} = & \left\{ \begin{array}{l}
+ \frac{\Delta t}{\Delta x}\delta_{1;h}^{n+1}F_{1;h+\frac{1}{2}}^{n} + \frac{\Delta t}{(\Delta x)^2}D_{1;h+\frac{1}{2}}^{n} - \frac{\Delta t}{\Delta x}\left(1-\delta_{1;h-1}^{n+1}\right)F_{1;h-\frac{1}{2}}^{n} + \frac{\Delta t}{(\Delta x)^2}D_{1;h-\frac{1}{2}}^{n} \\[2mm]
+ \frac{\Delta t}{\Delta x}\delta_{2;h}^{n+1}F_{2;h+\frac{1}{2}}^{n} + \frac{\Delta t}{(\Delta x)^2}D_{2;h+\frac{1}{2}}^{n} - \frac{\Delta t}{\Delta x}\left(1-\delta_{2;h-1}^{n+1}\right)F_{2;h-\frac{1}{2}}^{n} + \frac{\Delta t}{(\Delta x)^2}D_{2;h-\frac{1}{2}}^{n} \\[2mm]
+ \frac{\Delta t}{\Delta \alpha}\delta_{\alpha;k}^{n+1}F_{\alpha;k+\frac{1}{2}}^{n} + \frac{\Delta t}{(\Delta \alpha)^2}D_{\alpha;k+\frac{1}{2}}^{n} - \frac{\Delta t}{\Delta \alpha}\left(1-\delta_{\alpha;k-1}^{n+1}\right)F_{\alpha;k-\frac{1}{2}}^{n} + \frac{\Delta t}{(\Delta \alpha)^2}D_{\alpha;k-\frac{1}{2}}^{n} \\[2mm]
+ \frac{\Delta t}{\Delta \beta}\delta_{\beta;l}^{n+1}F_{\beta;l+\frac{1}{2}}^{n} + \frac{\Delta t}{(\Delta \beta)^2}D_{\beta;l+\frac{1}{2}}^{n} - \frac{\Delta t}{\Delta \beta}\left(1-\delta_{\beta;l-1}^{n+1}\right)F_{\beta;l-\frac{1}{2}}^{n} + \frac{\Delta t}{(\Delta \beta)^2}D_{\beta;l-\frac{1}{2}}^{n}
\end{array} \right\} P_{h,k,l}^{n+1} \\[3mm]
& + \left\{ \begin{array}{l}
\frac{\Delta t}{\Delta x}\left(1-\delta_{1;h}^{n+1}\right)F_{1;h+\frac{1}{2}}^{n} - \frac{\Delta t}{(\Delta x)^2}D_{1;h+\frac{1}{2}}^{n} \\[2mm]
+ \frac{\Delta t}{\Delta x}\left(1-\delta_{2;h}^{n+1}\right)F_{2;h+\frac{1}{2}}^{n} - \frac{\Delta t}{(\Delta x)^2}D_{2;h+\frac{1}{2}}^{n}
\end{array} \right\} P_{h+1,k,l}^{n+1} \\[3mm]
& + \left[ \frac{\Delta t}{\Delta \alpha}\left(1-\delta_{\alpha;k}^{n+1}\right)F_{\alpha;k+\frac{1}{2}}^{n} - \frac{\Delta t}{(\Delta \alpha)^2}D_{\alpha;k+\frac{1}{2}}^{n} \right] P_{h,k+1,l}^{n+1} \\[3mm]
& + \left[ \frac{\Delta t}{\Delta \beta}\left(1-\delta_{\beta;l}^{n+1}\right)F_{\beta;l+\frac{1}{2}}^{n} - \frac{\Delta t}{(\Delta \beta)^2}D_{\beta;l+\frac{1}{2}}^{n} \right] P_{h,k,l+1}^{n+1} \\[3mm]
& + \left\{ \begin{array}{l}
- \frac{\Delta t}{\Delta x}\delta_{1;h-1}^{n+1}F_{1;h-\frac{1}{2}}^{n} - \frac{\Delta t}{(\Delta x)^2}D_{1;h-\frac{1}{2}}^{n} \\[2mm]
- \frac{\Delta t}{\Delta x}\delta_{2;h-1}^{n+1}F_{2;h-\frac{1}{2}}^{n} - \frac{\Delta t}{(\Delta x)^2}D_{2;h-\frac{1}{2}}^{n}
\end{array} \right\} P_{h-1,k,l}^{n+1} \\[3mm]
& + \left[ - \frac{\Delta t}{\Delta \alpha}\delta_{\alpha;k-1}^{n+1}F_{\alpha;k-\frac{1}{2}}^{n} - \frac{\Delta t}{(\Delta \alpha)^2}D_{\alpha;k-\frac{1}{2}}^{n} \right] P_{h,k-1,l}^{n+1} \\[3mm]
& + \left[ - \frac{\Delta t}{\Delta \beta}\delta_{\beta;l-1}^{n+1}F_{\beta;l-\frac{1}{2}}^{n} - \frac{\Delta t}{(\Delta \beta)^2}D_{\beta;l-\frac{1}{2}}^{n} \right] P_{h,k,l-1}^{n+1}
\end{aligned} \tag{42}
$$

Equation (42) is the discretized version of the FPE that represents the proposed methodology of this study. A comparison of Eqs. (24) and (25) provides the expressions for the $F$ and $D$ parameters, while Eqs. (33) to (36) provide the expressions for the $\delta$ parameters, all of which would then complete the solution of Eq. (42). This equation has to be solved implicitly in order to compute the ensemble behavior and variability of a hydrologic system defined by the stochastic Saint-Venant equations, and it does that by solving for the joint PDF of the state variables within the $x$–$\alpha$–$\beta$ domain. Equation (42) provides a more efficient approach to solve for the ensemble behavior and variability of the stochastic unsteady open-channel flow in a rectangular, prismatic channel under uncertain roughness coefficient, by running only one simulation. This proposed FPE methodology can also be expanded to problems with uncertainties in other channel or flow parameters. The performance of the proposed FPE methodology is evaluated in a companion paper by Dib and Kavvas (2017) which compares its results to those obtained by the MC approach.

## 4 Summary and conclusions

This study proposed a new methodology to model the expected behavior and variability of a system described by the stochastic open-channel flow equations. The governing equations that were used to represent the flood routing problem




in this study are the continuity and momentum equations, otherwise known as the Saint-Venant equations. Many uncertainties can add to the complexity of solving the Saint-Venant equations in engineering routing problems. These uncertainties may include uncertainties in the channel's physical and geometric properties, as well as uncertainties in the lateral inflows and upstream boundary conditions, all of which render the Saint-Venant equations stochastic. As such, the dependent variables that will be solved for by these equations will also become stochastic, thus requiring that their statistical properties be solved for at specific time-space locations. Therefore, with uncertain parameters, the Saint-Venant equations have to be solved within a stochastic framework in order to quantify the ensemble behavior and variability of the system being considered. While the Mote Carlo method is a viable approach for the solution of such a stochastic unsteady open-channel flow problem, its computational expense and its large number of simulations act to its disadvantage. Hence, a new methodology was proposed in this study by which the statistical properties of the dependent variables of the considered hydrologic problem may be obtained in only one single simulation.

The proposed FPE methodology derived in this study involved upscaling the governing stochastic differential equations by developing their corresponding Lagrangian–Eulerian Fokker–Planck Equation (LEFPE), thus transforming the original stochastic equations into the framework of a deterministic differential equation. The deterministic LEFPE that describes the time-space evolution of the probability density function of the unsteady open-channel flow state variables, was developed following the method in Kavvas (2003) after the governing Saint Venant equations were transformed into their characteristic form by using the method of characteristics. Through simplifications, this LEFPE was reduced to a classical FPE that could be solved deterministically for the evolution of the probability density of the state variables of the system. The obtained FPE of the proposed methodology was discretized in an implicit manner following Chang and Cooper (1970) in order to obtain the equations that may be solved numerically to determine the ensemble behavior of the considered system.

Therefore, by solving a deterministic and linear FPE through only one simulation, the proposed methodology provides a more efficient approach to solve for the ensemble behavior and variability of a system described by the stochastic open-channel flow equations. In this study, the open-channel flow problem was considered for a rectangular, prismatic channel under an uncertain roughness coefficient. However, the proposed methodology can be expanded to problems which assume uncertainties that arise from other flow or channel parameters. The application and validation of this methodology is provided in a companion paper by Dib and Kavvas (2017), in which the statistical results of the proposed FPE methodology are compared against the results obtained by the MC approach.

## 5 Data availability

This study involved the theoretical development and derivation of a new proposed methodology for the stochastic solution of unsteady open-channel flow. No data was used in the derivation process.



*Competing interests*. The authors declare that they have no conflict of interest.

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
