# Peer review of "Ensemble modeling of stochastic unsteady open-channel flow in terms of its time-space evolutionary probability distribution: theoretical development"

_Hydrology and Earth System Sciences, 2017_

## Referee Comment (RC1) · Anonymous Referee #1 · 31 Aug 2017

General comments:

The authors derive PDF equations for the Saint-Venant equations, with the stochasticity stemming from an uncertain Manning's roughness coefficient. They discretise the PDF equation via a finite difference scheme. The discretised PDF equation is tested and analysed in the companion paper.

The topic of the paper is interesting and the derivations are concise, yet easy to follow. I believe that PDF methods will become important modelling tools for hydrological pro-

cesses. Nevertheless, there is room for improvements. For details, see the following specific comments.

Specific comments:

The authors repeat themselves more than once on specific topics and some parts are too detailed or even obvious. Some examples would be p.5 l. 2-3., p. 5, l. 14-26, or p. 9, l. 6-7.

If the PDF is obtained by a FPE, it implies that the PDF is approximated and completely determined by its first two moments. This is somewhat stated in the introduction, but not when introducing eq. (9). Please state this approximation explicitly. This restriction has to also be mentioned when comparing your method to MC approaches, from which any statistical moment can be derived, given enough particles and ensemble members.

You write that the FPE methodology is more efficient, because it can be calculated in one single simulation run. This is an invalid argument, as PDFs are highly dimensional functions and quickly become computationally unfeasible. The computational efficiency is to be proven, refer to the companion paper.

Can you give examples where neglecting the cross-covariance terms, like eq. (21), break down?

I know that Manning's equation is well established in the form you used in your work, but the conversion factor k is superfluous if the correct units are used consistently. I suggest that you consider taking k out.

How can the FPE methodology be expanded to problems with more uncertain parameters? Which parameters can be assume uncertain and how would the methodology have to be adapted?

Why did you chose the numerical scheme of Chang and Cooper? How does it compare to alternatives? Furthermore, please provide the reader with more information about the scheme, e.g. the accuracy, or the order of convergence.

In the outlook, please elaborate more on how to expand the methodology to problems with more or different sources of uncertainty.

Technical corrections:

Check the indentations at the beginning of chapters and after equations, delete them.

In equation (9), you use a semi-colon to separate the two arguments of the covariance function, but a comma in equation (10), be consistent.

When referring to Kavvas 2003 for equation (9), also state the equation number.

On p. 5., in l. 18 you state that finite difference schemes are defined on fixed rectangular x-t grid. But these schemes can be used on adaptive grids too.

The angular brackets in eq. (10) have different sizes, chose one size.

The arguments of a PDF are usually separated by a semi-colon into arguments for which the PDF is a density and normal arguments, e.g. Pope (1985).

Maybe eq. (19) can be written in vector notation for readability. Although I am not sure how much improvement this will bring.

Convection is a rather fuzzy term. You should stay with the mathematical rigorosly defined terms, like advection, diffusion, or transport terms. Please correct the terms on page 11.

On p. 13, l. 10 you write about the conservation of particles, although no particles where introduced in your paper. Reformulate this part. For example, you could write about the conservation of the normalisation.

For the weighting factors, you used the variable delta, which is already used for the delta function, please chose a different variable.

I suggest you remove equations (30) - (32), (34) - (36), and (38) - (40) and simply write "analogue for the other dimensions".

---

## Referee Comment (RC2) · Anonymous Referee #2 · 12 Sep 2017

In this two-paper series the authors present a model of the probability distribution of relevant hydraulic quantyties,i.e., water depth and water discharge, along a regular (rectangular) channel under transient flow conditions and for a random roughness (Manning) coefficient. Flow is assumed one-dimensional and governed by the classical de Saint Venant equation with a spatially uniform and normally distributed Manning coefficient. The contribution is splitted into two manuscripts, the first one reporting the derivation of a differential equation for the PDF, under the form of the Fokker-Plank Equation (FPE), and the second presenting an illustrative example. The first consideration I have, after reading carefully the two manuscripts, is that this work would be better communicated if merged into a single contribution. The first manuscript does not stand alone because the theoretical derivations are not novel, but an application of a theory already presented in a previous publication by Kavvas (2003). Also the numerical technique proposed to solve the FPE equation is not new, rather an adaptation of the numerical scheme proposed by Chang and Cooper (1970). On the other hand, the second manuscript describes an idealized application with a regular rectangular channel and a triangular hydrograph at the upstream section. In my view, these two manuscripts can be easily merged into one, by transferring some long expressions of the first manuscreipt into an appendix and removing the first 5 pages (out of 13) of the second manuscript, which summarize what was presented in the first manuscript.

In the first manuscript the authors present a general form of the differential equation for the PDF (Eq. 19), based on theoretical developments presented in a previous paper (Kavvas, 2003. This expression contains autocovariances and cross-covariances of the state stochastic variables and is further simplified by introducing a number of assumptions as expressed by Eqs (20) and (22). These assumptions are in part supported by a previous study (Eq. 20), but some of them have been introduced without justification, other than mathematical convenience. The validation of these hypotheses is left to the comparison with Monte Carlo simulations, which is presented in the second manuscript. This further suggests the opportunity to merge the two manuscripts into one. Assuming that sufficient justification for the the assumptions of Eq. (20) can be found in a previous paper, as declared by the authors at page 11, the assumption (22) seem rather extreme since it implies zero correlation (and autocorrelation) between the stochastic variables for time lags larger than 0. This looks a rather strong assumption considering the diffusive nature of the De Saint Venant equation. In other words, this hypothesis implies that $\alpha$ and $\beta$ (Eqs. 13 and 14), for example, are two independent white noises. Given the expressions (13) and (14) this hypothesis translates to both velocity $V$ and the celerity $c$, which become white noise as well, while one expects these quantities to be correlated, and cross-correlated. More convincing

arguments are needed here than the simple hypothesis, not supported by evidences, that the stochastic variables have short memory (page 11, line 17).

---

## Author Comment (AC1) · 1 Nov 2017

**Response to the comments of Anonymous Referee #1 published on August 31, 2017 concerning the manuscript with reference number: hess-2017-393.**

We would like to thank Referee #1 for his/her insightful comments. Our responses to the specific points raised by the referee are provided below. Please note that the referee's comments will be presented in italics, preceded by a "**C**", while the corresponding authors' responses will be presented in normal typeface with a blue font, preceded by an "**R**". Please note that the pages and line numbers provided in the referee comments are from the original version of the manuscript, whereas any pages and line numbers mentioned in the authors' responses correspond to a revised version of the manuscript, unless stated otherwise. The revised version of the manuscript has been placed at the end of this document and is a result of revisions made following comments from both Referee #1 and Referee #2. The same revised version has also been placed at the end of the responses to the comments of Referee #2.

**General Comments**

**C1:** *The authors derive PDF equations for the Saint-Venant equations, with the stochasticity stemming from an uncertain Manning's roughness coefficient. They discretise the PDF equation via a finite difference scheme. The discretised PDF equation is tested and analysed in the companion paper. The topic of the paper is interesting and the derivations are concise, yet easy to follow. I believe that PDF methods will become important modelling tools for hydrological processes. Nevertheless, there is room for improvements. For details, see the following specific comments.*

**R1:** We would like to thank the referee for this review.

**Specific Comments**

**C2:** *The authors repeat themselves more than once on specific topics and some parts are too detailed or even obvious. Some examples would be p.5 l. 2-3., p. 5, l. 14-26, or p. 9, l. 6-7.*

**R2:** We understand the view of the referee regarding this point. We did not intend for there to be any repetition or excessive details that would cause the reading to be arduous. Our goal was to make this paper as easy to understand and as accessible as possible to the readers, especially those interested in applying the methodology explained in the paper. However, we worked on adjusting the portions indicated by the referee in order to remove any repetition or excessive obvious details, as shown below.

Page 5: Lines 2-3 in the original manuscript: The sentence spanning Lines 2-3 in the original version of the manuscript "It should be noted … derivation." was deleted.

Page 5: Lines 14-26 in the original manuscript: These two paragraphs were summarized further, and reduced to the portion shown on Page 5: Lines 12-20 of the revised manuscript.

Page 9: Lines 6-7 in the original manuscript: Unneeded details have been removed and the new sentence can be found on Page 9: Line 3 of the revised manuscript.

After going through the whole manuscript again, a few other adjustments were done to other parts of the manuscript, by following the referee's point above.

**C3:** *If the PDF is obtained by a FPE, it implies that the PDF is approximated and completely determined by its first two moments. This is somewhat stated in the introduction, but not when introducing eq. (9). Please state this approximation explicitly. This restriction has to also be mentioned when comparing your method to MC approaches, from which any statistical moment can be derived, given enough particles and ensemble members.*

**R3:** The referee's point has been made clearer to the reader with a few text additions before introducing Eq. (9) (Page 7: Lines 18-20), and again very briefly prior to presenting Eq. (19) (Page 9: Line 14). We will also include such clarifications in the companion paper, especially during the comparison of our method to the MC approach, as the referee has suggested.

**C4:** *You write that the FPE methodology is more efficient, because it can be calculated in one single simulation run. This is an invalid argument, as PDFs are highly dimensional functions and quickly become computationally unfeasible. The computational efficiency is to be proven, refer to the companion paper.*

**R4:** As pointed out by the referee, the mention of the efficiency of the proposed methodology in this manuscript was adjusted, specifically on Page 17: Line 6 (the mention that was on Page 17: Line 33 has been removed due to a rearrangement of ideas in the final two paragraphs of the conclusion). While we acknowledge the referee's comments regarding this point, we note that the computational efficiency of this proposed methodology may truly stand out in situations which involve dealing with several uncertainties simultaneously, including a few uncertain parameters as well as uncertain upstream boundary conditions. In this case, the Monte Carlo approach may require a much larger number of simulations in order to obtain results with the same required level of accuracy, which would exponentially add to the computational expense of the method. Whereas, the FPE methodology would still require solving the same FPE with a few additional terms (as explained in **R7** below) without significantly affecting the required computations.

**C5:** *Can you give examples where neglecting the cross-covariance terms, like eq. (21), break down?*

**R5:** As is mentioned in the manuscript, the cross-covariance terms between any two different $\eta$ functions [e.g., Eq. (21)] can be assumed to be negligible because they have been shown, in Liang and Kavvas (2008), to be much smaller in magnitude than the autocovariance of one $\eta$ function. That would be the situation in most cases, especially where the $\eta$ functions are unrelated, or not closely related to each other. However, when the cross-covariance is being computed for two $\eta$ functions that are related or similar to each other, such an assumption may break down. Such a similarity may either come from the similarities of the functions to each other (in regard to their expressions), the similarity of the behavior of these functions (e.g., both can be associated with the same time-series model), or even from the existence of periodicity within both $\eta$ functions with frequencies that are not too far off from each other. All such examples may cause the cross-covariance terms of the two $\eta$ functions to be closer in value to the autocovariance values, possibly making it no longer valid to neglect the cross-covariance terms.

This has been added, in a summarized manner, to Page 11: Lines 16-18.

**C6:** *I know that Manning's equation is well established in the form you used in your work, but the conversion factor k is superfluous if the correct units are used consistently. I suggest that you consider taking k out.*

**R6:** We have considered the referee's suggestion. However, the conversion factor (k) is present in several of the complex expressions in the equations of this manuscript, and in some cases, it may not be easily linked back to the rest of Manning's equation. As such, we believe that it would be preferable to keep the conversion factor in the equations for the best interest and the ease-of-use of the reader, especially if the reader would be interested in replicating the study or using the methodology, regardless of the units that the reader may choose.

**C7:** *How can the FPE methodology be expanded to problems with more uncertain parameters? Which parameters can be assumed uncertain and how would the methodology have to be adapted?*

**R7:** [Equations provided in this response refer to the original manuscript, unless otherwise noted.]

In the second paragraph of the introduction, we list several possible uncertainties that may affect problems involving open channel flows. Among them, we mention uncertainties arising from channel geometric parameters (e.g., bed slope and cross section geometry), lateral inflows/outflows, and upstream boundary conditions. Examples of translating those uncertainties into the equations of this study will be provided in a simplified manner below.

If we assume that the channel bed slope is uncertain, then $S_0$ can be counted as an uncertain parameter. This change can be incorporated by including $S_0$ into the expectation expressions and the variance expressions of Eq. (24), thus changing the expressions of the advection and diffusion coefficients in the $\alpha$ and $\beta$ directions. A similar case arises if the channel width ($b$) is considered uncertain, in which case it must also be included into the expectation and variance expressions of Eq. (24). However, in cases where the channel width is assumed to be changing with position, or even when the channel cross section geometry is assumed uncertain with position, the continuity and momentum equations will have to be adjusted accordingly, and the resulting changes will subsequently alter the expressions in the equations that follow. Similarly, in the case that lateral inflows/outflows exist and are uncertain, additional terms corresponding to the lateral inflows/outflows will have to be added to the continuity and momentum equations, and subsequently to Eqs. (4), (6), (16), (18), and (24). Again, with these additional terms being uncertain, they will have to be included into the expectation and variance expressions of Eq. (24). Finally, in the case that the upstream boundary conditions are uncertain, then the upstream boundary conditions used for solving the problem will change from being deterministic to being stochastic, represented by some PDF at the upstream boundary that defines the uncertainty in those conditions.

Note that all of the uncertainties mentioned above may be incorporated at the same time in the FPE methodology by simultaneously applying all of the above changes.

A summarized version of the above discussion is provided in the conclusion, as requested in **C9** below.

**C8:** *Why did you chose the numerical scheme of Chang and Cooper? How does it compare to alternatives? Furthermore, please provide the reader with more information about the scheme, e.g. the accuracy, or the order of convergence.*

**R8:** In this study, the proposed methodology involves a classical FPE that directly solves for the probability density of the dependent variables. In general, finite difference and finite elements methods, among others, have been widely used to solve FPEs numerically. Many studies have compared several such methods to determine how they perform against each other in solving different FPEs. In their study, Pichler et al. (2011) looked at the central finite difference method, the alternating directions implicit (ADI) method, as well as finite element methods. They mentioned that finite difference methods are computationally more economical than finite element methods as the number of dimensions increases. Park and Petrosian (1996) performed a comparison between implicit schemes (including a fully implicit mid-point difference method), the schemes presented in Larsen et al. (1985), as well as the Chang and Cooper (1970) scheme; they also studied the semi-implicit forms of these schemes (e.g., the Crank-Nicholson scheme). They concluded that, among these schemes, the best finite difference method for solving their FPEs was the Chang and Cooper method, as it was the most robust and most stable over a wide range of parameters among the other methods tested in their study.

In fact, the Chang and Cooper scheme has been cited as one of the most widely known schemes for solving the classical FPE numerically. An important advantage of this scheme is that it satisfies two crucial requirements regarding solving for the probability density in a FPE: the requirement that the probability density cannot be negative, and that the probability mass must be conserved inside the system. The differencing scheme of Chang and Cooper formulates a generalized flux of probability mass that results in a method with first-order convergence in space and time. While Chang and Cooper applied their numerical scheme to a one-dimensional FPE, it was successfully applied to two-dimensional FPEs in studies such as Kim et al. (2005a, b) and Cayar and Kavvas (2009a, b), which dealt with unsaturated water flow, root-water uptake, as well as unconfined groundwater flow. These studies have all been able to apply the Kavvas (2003) methodology and then to successfully solve the resulting two-dimensional FPEs by using the Chang and Cooper scheme. Consequently, following the information provided above, it was deemed appropriate to select and generalize the numerical scheme of Chang and Cooper for the solution of the multi-dimensional FPE obtained in this study.

Please note that a similar version to the above discussion has been added at the beginning of Section 3.2 which tackles the numerical solution for the FPE; this can be found on Page 14: Lines 7-19. Moreover, following the referee's suggestion, we added some more information regarding the Chang-Cooper scheme in the manuscript (Page 14: Lines 20-22).

**C9:** *In the outlook, please elaborate more on how to expand the methodology to problems with more or different sources of uncertainty.*

**R9:** For a detailed explanation regarding such an expansion of the methodology, please refer to response **R7**. Following the referee's comment **C9**, a summarized version of **R7** was added to the end of the conclusion to elaborate on how to expand the methodology to problems with other or additional uncertainty sources. Please refer to Page 18: Line 27 – Page 19: Line 3 to view the summarized explanation added to the manuscript.

**Technical corrections**

**C10:** *Check the indentations at the beginning of chapters and after equations, delete them.*

**R10:** Indentations have been deleted from beginning of chapters and after equations.

**C11:** *In equation (9), you use a semi-colon to separate the two arguments of the covariance function, but a comma in equation (10), be consistent.*

**R11:** The comma has been substituted by a semi-colon in Eq. (10) to be consistent with Eq. (9) and with all the other equations in the manuscript.

**C12:** *When referring to Kavvas 2003 for equation (9), also state the equation number.*

**R12**: The equation number from Kavvas 2003 has been added (Page 7: Line 18).

**C13:** *On p. 5., in l. 18 you state that finite difference schemes are defined on fixed rectangular x-t grid. But these schemes can be used on adaptive grids too.*

**R13:** The sentence has been adjusted following the referee's comment (Page 5: Line 15).

**C14:** *The angular brackets in eq. (10) have different sizes, chose one size.*

**R14:** The size of the angular brackets has been corrected in equation (10).

**C15:** *The arguments of a PDF are usually separated by a semi-colon into arguments for which the PDF is a density and normal arguments, e.g. Pope (1985).*

**R15:** This has been corrected in all the respective equations following the referee's comment.

**C16:** *Maybe eq. (19) can be written in vector notation for readability. Although I am not sure how much improvement this will bring.*

**R16:** Writing Eq. (19) in vector notation would most likely provide an expression that would be very similar to Eq. (9). On the other hand, leaving Eq. (19) in its current, expanded form provides the reader with a clearer visualization of all the different advection and diffusion terms corresponding to each of the four dimensions related to the problem being solved. This may be more helpful to the reader for following through the explanations and derivations in the remainder of the manuscript, and for use when replicating the problem. As such, we believe that it would be preferable to keep Eq. (19) in its current expanded form.

**C17:** *Convection is a rather fuzzy term. You should stay with the mathematical rigorously defined terms, like advection, diffusion, or transport terms. Please correct the terms on page 11.*

**R17:** The terms "convection" have been substituted by "advection" as suggested by the referee.

**C18:** *On p. 13, l. 10 you write about the conservation of particles, although no particles where introduced in your paper. Reformulate this part. For example, you could write about the conservation of the normalisation.*

**R18:** It is true that no discussion of particles in a specific manner occurs in any other part of the manuscript, and we thank the referee for this input regarding this point. We note that the state variable in this study exists in the form of probability density, and so, in this case the probability mass in the system has to be conserved in any evolution time step when modelling it. As such, and following the referee's suggestion, the phrase in question has been changed from "conservation of the particles" to "conservation of the probability mass" (Page 14: Line 23).

**C19:** *For the weighting factors, you used the variable delta, which is already used for the delta function, please chose a different variable.*

**R19:** The weighting factors have been all changed from $\delta$ to $\lambda$ in the equations and in the text.

**C20:** *I suggest you remove equations (30) - (32), (34) - (36), and (38) - (40) and simply write "analogue for the other dimensions".*

**R20:** All the equations mentioned by the referee have been removed as suggested, and expressions similar to the suggested expression "analogue for the other dimensions" have been included. Also, wherever deemed necessary, slight modifications have been made to the text portions that are related to these equations in order to cope with the deletion of the equations.

**References**

[revised manuscript text omitted]

---

## Author Comment (AC2) · 1 Nov 2017

**Response to the comments of Anonymous Referee #2 published on September 12, 2017 concerning the manuscript with reference number: hess-2017-393.**

We would like to thank Referee #2 for his/her insightful feedback. Our responses to the specific points raised by the referee are provided below. Please note that the referee's comments will be presented in italics, preceded by a "**C**", while the corresponding authors' responses will be presented in normal typeface with a blue font, preceded by an "**R**". Please note that the pages and line numbers provided in the referee comments are from the original version of the manuscript, whereas any pages and line numbers mentioned in the authors' responses correspond to a revised version of the manuscript, unless stated otherwise. The revised version of the manuscript has been placed at the end of this document and is a result of revisions made following comments from both Referee #2 and Referee #1. The same revised version has also been placed at the end of the responses to the comments of Referee #1.

**C1:** *In this two-paper series the authors present a model of the probability distribution of relevant hydraulic quantyties,i.e., water depth and water discharge, along a regular (rectangular) channel under transient flow conditions and for a random roughness (Manning) coefficient. Flow is assumed one-dimensional and governed by the classical de Saint Venant equation with a spatially uniform and normally distributed Manning coefficient. The contribution is splitted into two manuscripts, the first one reporting the derivation of a differential equation for the PDF, under the form of the Fokker-Plank Equation (FPE), and the second presenting an illustrative example.*

**R1:** We would like to thank the referee for this review.

**C2:** *The first consideration I have, after reading carefully the two manuscripts, is that this work would be better communicated if merged into a single contribution. The first manuscript does not stand alone because the theoretical derivations are not novel, but an application of a theory already presented in a previous publication by Kavvas (2003). Also the numerical technique proposed to solve the FPE equation is not new, rather an adaptation of the numerical scheme proposed by Chang and Cooper (1970). On the other hand, the second manuscript describes an idealized application with a regular rectangular channel and a triangular hydrograph at the upstream section. In my view, these two manuscripts can be easily merged into one, by transferring some long expressions of the first manuscreipt into an appendix and removing the first 5 pages (out of 13) of the second manuscript, which summarize what was presented in the first manuscript.*

**R2:** We thank the referee for the comment. The ensemble-averaging technique used in this study was indeed developed by Kavvas (2003) and this technique has been applied to other processes with different governing equations where Fokker–Planck Equations (FPEs) specific to those processes were obtained and applied successfully. However, this technique had never been applied to the Saint-Venant equations to tackle the stochastic unsteady open-channel flow problem. As such, the novelty of the proposed FPE methodology that was developed in this manuscript was, firstly, to figure out how to apply the Kavvas (2003) technique to the Saint-Venant equations especially through the transformations that provided us with the state variables α and β and that allowed us to write the Saint-Venant equations as four Ordinary Difference Equations (ODEs) in the specific forms of Eqs. (15) to (18) (for this was not a straightforward process), and secondly, to go forth with developing the FPE that is specifically for the stochastic unsteady open-channel flow process, an equation which has not been developed before. Hence, this study clearly derives and presents an entirely new FPE that can be used to solve for

the probability density of the state variables of a stochastic open-channel system, which is not found elsewhere in the literature. And while the numerical discretization was made following the Chang and Cooper (1970) scheme, the scheme was generalized from its original one-dimensional form and adapted to the four-dimensional FPE that was being solved in this study. Therefore, joining this manuscript and the companion manuscript into one manuscript, while placing a large portion of this first manuscript in the appendix, would mostly take away from the importance of these equations and from the work that was done in arriving to those equations. As such, we believe in the novelty of the equations derived in this manuscript, and thus we believe in its ability to stand as its own manuscript. We would also like to note that Referee #1, who has read and reviewed both manuscripts, has not mentioned any desire for the joining of the two papers into one, in which case we would assume that Referee #1 may not have seen any concern with them being two standalone, companion papers.

**C3:** *In the first manuscript the authors present a general form of the differential equation for the PDF (Eq. 19), based on theoretical developments presented in a previous paper (Kavvas, 2003. This expression contains autocovariances and cross-covariances of the state stochastic variables and is further simplified by introducing a number of assumptions as expressed by Eqs (20) and (22). These assumptions are in part supported by a previous study (Eq. 20), but some of them have been introduced without justification, other than mathematical convenience. The validation of these hypotheses is left to the comparison with Monte Carlo simulations, which is presented in the second manuscript. This further suggests the opportunity to merge the two manuscripts into one. Assuming that sufficient justification for the the assumptions of Eq. (20) can be found in a previous paper, as declared by the authors at page 11, the assumption (22) seem rather extreme since it implies zero correlation (and autocorrelation) between the stochastic variables for time lags larger than 0. This looks a rather strong assumption considering the diffusive nature of the De Saint Venant equation. In other words, this hypothesis implies that α and β (Eqs. 13 and 14), for example, are two independent white noises. Given the expressions (13) and (14) this hypothesis translates to both velocity V and the celerity c, which become white noise as well, while one expects these quantities to be correlated, and cross-correlated. More convincing arguments are needed here than the simple hypothesis, not supported by evidences, that the stochastic variables have short memory (page 11, line 17).*

**R3:** [Equations provided in this response refer to the original manuscript, unless otherwise noted.]

One reason for the use of such an approximation as shown in Eq. (22) is a physical reason involving the representations of α and β, as well as the $\eta$ functions. In fact, as the referee mentioned, α and β are indeed functions of V and c, which also cause the $\eta$'s to be functions of V and c as well. But it is important to recall that the $\eta$ functions describe motions occurring in opposite directions, in a similar manner to how α may describe a forward propagation motion and β may describe a backward propagation motion, based on their equations. In such a case, we are modeling two different and opposing propagation directions. However, having opposite directions, the backward and forward motions will be expected to be weakly correlated. This is because the V-c expression in the forward direction will be expected to have very little correlation with the V-c expression in the backward direction. Hence, Eq. (22) would be an appropriate approximation of a such a very weak correlation. Furthermore, the order of magnitude analysis that was obtained in the Liang and Kavvas (2008) paper corroborates the use of such an approximation due to the higher order of magnitude of the variances relative to the autocovariances. In fact, it should be noted that in case the approximation of Eq. (22) was not used in this study, it would be expected that the numerical calculations would spontaneously reveal

that the autocovariance values will be negligible relative to the variance values for the expressions in question, which is what our approximation eventually specifies.

Moreover, we would like to note that this approximation of Eq. (22) is only made to the second group of 4 terms shown in Eq. (19). This is clear when observing Eq. (22) of the revised manuscript, which is an additional equation that was added to the revised manuscript. This is important because the resulting equation that we obtain [i.e., Eq. (23) in the original manuscript] is still a diffusive equation, which accounts for the second order effects in both space and time. The time-space varying covariances that were simplified with the approximation are reduced to time-space varying variances, but we are still able to represent the transience in the behavior of the process, not only in the first moment but also in the second moment, as we can see in the results of the companion paper. As such, we would like to stress that such an approximation did not prevent the representation of the diffusive nature of the Saint-Venant equation; this diffusive nature is still represented in Eq. (23). Finally, if this approximation were not to work, we were indeed planning to entertain a richer correlation structure for these covariance terms. However, since this simple approximation was acceptable for the results we obtained, we felt it was satisfactory to use it due to its convenience for the numerical manipulation involved.

Please note that an additional intermediate equation has been added between Eq. (21) and (22) of the original manuscript. This equation is Eq. (22) in the revised manuscript, and is followed by a summarized version of the above discussion (Page 12: Lines 2-7) to provide additional reasoning to the reader for the use of the approximation discussed above.

**References**

[revised manuscript text omitted]